# The Role of Off-Job Crafting in Burnout Prevention during COVID-19 Crisis: A Longitudinal Study

**DOI:** 10.3390/ijerph19042146

**Published:** 2022-02-14

**Authors:** Roald Pijpker, Philipp Kerksieck, Martin Tušl, Jessica de Bloom, Rebecca Brauchli, Georg F. Bauer

**Affiliations:** 1Health and Society/Rural Sociology, Wageningen University, 6706 KN Wageningen, The Netherlands; 2Public and Organizational Health, Center of Salutogenesis, Epidemiology, Biostatistics and Prevention Institute, University of Zurich, 8001 Zurich, Switzerland; philipp.kerksieck@uzh.ch (P.K.); martin.tusl@uzh.ch (M.T.); rebecca.brauchli2@zhaw.ch (R.B.); georg.bauer@uzh.ch (G.F.B.); 3Faculty of Social Sciences, Psychology, Tampere University, 33014 Tampere, Finland; j.de.bloom@rug.nl; 4Faculty of Economics and Business, University of Groningen, 9747 AJ Groningen, The Netherlands

**Keywords:** burnout, off-job crafting, COVID-19, longitudinal, employees, DRAMMA, prevention, Germany, Switzerland, pre-post COVID-19 study

## Abstract

The COVID-19 pandemic and remote working challenge employees’ possibilities to recover from work during their off-job time. We examined the relationship between off-job crafting and burnout across the COVID-19 crisis. We used a longitudinal research design, comprising one wave collected before the onset of the pandemic, in March 2019 (T1), and one wave collected during the first lockdown of the crisis in April 2020 (T2). We measured the six off-job crafting dimensions (Crafting for Detachment, Relaxation, Autonomy, Mastery, Meaning, and Affiliation) and burnout (fatigue/exhaustion) via a questionnaire among German and Swiss employees (*N* = 658; Age *M* = 47; 55% male). We found that both burnout levels and crafting for affiliation significantly decreased at T2 compared to T1. All off-job crafting dimensions and burnout correlated negatively cross-sectionally and longitudinally. Regression analyses showed that employees who crafted in their off-job time before and during the crisis experienced fewer burnout complaints during the crisis. Looking more closely at the subdimensions of off-job crafting, employees who crafted for detachment before and during, and for affiliation before the crisis, reported less burnout during the crisis. We conclude that off-job crafting may act as a buffer mechanism against burnout during the COVID-19 crisis.

## 1. Introduction

The continuing global COVID-19 pandemic and the related control measures seriously challenge employees’ ability to maintain their health while staying productive at work [1,2]. The pandemic affects employee health through direct pathways (e.g., via the fear of being affected by the virus in high-exposure occupations) and indirect pathways (e.g., via the fear of being economically affected by the measures to control the pandemic), resulting in mental health complaints and stress symptoms [3]. Moreover, a report published by Eurofound [4] showed that even employees who have experienced some improvements in their work situation after lifting the first lockdown (e.g., getting back to full working hours) still report high levels of work-related stress. In addition, research on teachers, a group frequently studied in burnout research, revealed that perceived threats due to work-related changes during the COVID-19 pandemic were related to higher burnout levels [5]. Besides its markedly adverse effects, the ongoing COVID-19 pandemic offers an excellent opportunity to increase our understanding of how employees recover from work stress in their non-working time before and during crisis situations, which may buffer the effects of the crisis on burnout complaints—a key predictor of adverse workplace health and well-being outcomes [6,7,8,9,10].

Based on a longitudinal study design, comprising one wave collected before the onset of the pandemic and one wave collected during the crisis, the present study examines the role of off-job crafting for burnout prevention in times of the COVID-19 pandemic. Insights into the potential role of off-job crafting may support the work of health-promotion practitioners and policymakers in enabling the workforce to develop and maintain optimal levels of health and well-being during public health crises situations.

### 1.1. Off-Job Crafting for Better Recovery from Work

Adaptive strategies to cope with highly demanding work can be classified into three types [11,12]: (a) dealing with depleted resources (e.g., coping strategies, recovery); (b) altering job characteristics (e.g., job crafting); and (c) work and non-work boundary management (e.g., segmentation). Off-job crafting belongs to the first category as it enables restoration of depleted resources during non-working time and employees’ capacity to cope with workplace stressors successfully, thereby preventing negative effects of job demands on employee burnout. Off-job crafting refers to employees’ proactive and self-initiated changes in their non-working lives to satisfy their psychological needs [13,14]. According to the identity-based integrative needs model of crafting, the satisfaction of psychological needs is understood as the core driver and result of crafting [15]. The needs addressed in off-job crafting are defined by the DRAMMA model [14], an acronym that stands for the six needs (Detachment, Relaxation, Autonomy, Mastery, Meaning, and Affiliation). Engaging in activities that lead to the satisfaction of these needs during non-working time may also alleviate burnout complaints. Based on Kujanpää et al. [13] and the groundbreaking work by recovery researchers such as Sonnentag and Fritz [16], *detachment* is defined as “switching off” from one’s thoughts related to work and tasks during off-job time. Aligning with the stressor–detachment model, crafting for psychological detachment such as purposefully “refraining from job-related activities and thoughts during non-work time” can reduce the effect of job stressors during non-work time [17]. Off-job crafting for *relaxation* encompasses proactively striving for feeling physically well and for reducing effortful activities. It has been found that strain, representing a lack of detachment and relaxation from work, predicted higher levels of burnout and lower life satisfaction [13,17]. Off-job crafting for *autonomy* reflects striving for a feeling of being in control over one’s actions, life, and choices. Off-job crafting for *mastery* refers to behaviors aiming for feelings of proficiency and skillfulness in the task in which employees engage, such as taking up pleasant challenges and learning opportunities. Off-job crafting for *meaning* taps into striving for the experience of a sense of purpose and significance in one’s life and activities. Finally, off-job crafting for affiliation refers to aiming for the experience of being closely related and emotionally connected to others.

Before the onset of the COVID-19 crisis, studies showed that the recovery experience of detachment, relaxation, and mastery were negatively associated with burnout [16]. More recently and related to the previous study, research indicated that all six DRAMMA dimensions had strong, negative associations with work-related stress [13]. Another study indicated negative associations of the general off-job crafting factor with burnout over a three-month time period [16,18]. These effects are indicative of spillover effects between life domains [19] when people craft during leisure time, and this can have an impact on well-being at work. Overall, earlier studies support the notion that off-job crafting could be a helpful and relevant proactive strategy for preventing burnout under changing living and working conditions such as the COVID-19 pandemic. Employees may proactively craft their off-job time when not demanded by work [20], and experience positive spillover in terms of better detachment and recovery, which prevents them from developing burnout symptoms in the long term. We therefore assume that off-job crafting can potentially reduce exhaustion and fatigue, as covered in the burnout concept.

### 1.2. Burnout

Following Kristensen et al. [21], we define burnout as the degree of physical and psychological fatigue and exhaustion that is perceived by the person concerning their work. In everyday working life, it is well established that burnout is closely associated with various work-related and non-work-related health and well-being outcomes [6,7,8,9,10]. For example, burnout leads to adverse physical (e.g., musculoskeletal pain, severe injuries, type 2 diabetes), psychological (e.g., insomnia, depressive feelings, anxiety), and occupational (e.g., high sick-leave costs, lower job performance, job dissatisfaction) consequences [22]. Burnout is often the result of high job demands, in particular role stress, stressful events, role ambiguity, role conflict, and work pressure [23,24]. These demands are particularly harmful when job and personal resources, such as social support, autonomy, and self-efficacy are lacking [25,26]. Consequently, employees become chronically exhausted and psychologically distance themselves from their work, thereby impairing feelings of meaningfulness and the fulfilment of inherent psychological needs [27]. Since the onset of the pandemic, the role of employees’ non-working lives plays an important role in maintaining their (workplace) health and well-being. For example, a recent cross-sectional study conducted by Tusl et al. [2] showed that employees who experienced the changes after the pandemic hit as positive, such as having more leisure time, reported higher levels of mental well-being and self-related health than employees who experienced these changes as negative.

Besides the important role of reducing job demands and strengthening job resources in burnout prevention in general [28], employees are not passive agents undergoing a crisis. They can employ adaptive regulation strategies to prevent or diminish the onset of burnout complaints [29]. Although adaptive strategies for burnout prevention have received more attention recently within employees’ working life [28], little research exists for such proactive strategies within non-working life [29]. Crafting the non-work life domains is a promising stream of research still in its infancy [29,30]. It is therefore important to explore the role of off-job crafting as a possible pathway for alleviating burnout complaints in crisis situations. 

### 1.3. Study Aims and Research Questions

In an effort to expand prior studies, the present study aims to understand (1) the impact of the COVID-19 crisis on off-job crafting and burnout, and (2) to examine cross-sectional and longitudinal relationships between off-job crafting and burnout before and during the pandemic. Thereby, this study provides new insights into the extent to which people proactively craft their non-working life in a way that potentially protects against burnout. Since this study draws on longitudinal data collected before (T1) and during the crisis (T2), knowledge will be gained on how employees cope during and beyond crisis situations via off-job crafting, which is also little understood. Therefore, we focus on the following explorative research question:

Research Question 1: To what extent do both burnout and off-job crafting change during the COVID-19 crisis (T2) compared to before the crisis (T1)?

As the pandemic is unprecedented in modern working life and the impact of telework on a massive scale has neither been investigated with regard to burnout nor off-job crafting, this research question is explorative. Earlier research has shown that burnout levels can rise during a crisis due to, for example, increasing levels of job demands, lowering levels of job resources, and rising levels of job insecurity [31]. On the other hand, telework may also provide resources to people such as enhanced autonomy, social support and increased self-discipline [32]. In a drastically changing work and private life situation, crafting may be required in order to cope with these new challenges. However, people may also struggle and feel that they lack the personal resources to invest in crafting efforts, which may reduce actual crafting behaviors, similar to the recovery paradox [33]. The recovery paradox describes a situation in which workers find it particularly hard to recover during times of high job demands and work stress—a situation in which recovery is most needed.

Regarding the relationship between off-job crafting (i.e., crafting for DRAMMA) and burnout, based on the above literature review, we propose the following explorative research question:

Research Question 2: To what extent is off-job crafting (i.e., the total concept and the six subdimensions) related to burnout?

(2a) Cross-sectionally (i.e., How is off-job crafting at T1 and T2 associated with burnout at T1 and at T2, respectively?);

(2b) Longitudinally (i.e., How is off-job crafting at T1 associated with burnout at T2?)

Finally, we explore whether off-job crafting before and during the crisis can predict changes in burnout during the crisis and how off-job crafting during the crisis is related to burnout during the crisis:

Research Question 3a: To what extent is off-job crafting (i.e., the total concept and the six subdimensions) before the COVID-19 crisis (T1) related to a change in burnout during the crisis (burnout at T2 controlled for burnout and off-job crafting at T1)?

Research Question 3b: To what extent is off-job crafting (i.e., the total concept and the six subdimensions) during the COVID-19 crisis (T2) related to burnout during the crisis (burnout at T2 controlled for burnout at T1)?

Following earlier research on the beneficial effects of crafting and its importance for recovery, we expect that off-job crafting can act as a buffer that can prevent increasing burnout complaints during the COVID-19 crisis.

## 2. Methods

### 2.1. Participants and Procedures

A prospective longitudinal design was employed among employees in Germany and Switzerland, comprising two waves of measurements with a one-year time interval between the waves. As a baseline to compare the situation before the outbreak of COVID-19, we use a wave collected in March 2019, constituting the baseline measure (T1). The second wave was collected during the first lockdown in April 2020 to reflect the COVID-19 crisis (T2). Participants were recruited via Respondi (respondi.com), which is a high-quality panel provider based in Germany. Hence, the data allow us to follow the same participants throughout both waves, covering up to 12 months before the COVID-19 outbreak (T1) and the peak of the COVID-19 lockdown measures (T2). Only employees within the age range of 18 to 65 years who worked more than 20 h per week were included, excluding self-employed people. Participants were from a range of occupational sectors, including the health and social sector, public sector, sales, agriculture, production of goods, information and communication, finance, research, education, hospitality, transport, and construction.

The full sample collected in the first wave included *N* = 1501 participants, of which those who participated in the second wave *N* = 658 participants (44% participation). Men were more likely to participate in both waves, with 55% of those who participated in both waves being male compared to 51% of those who participated in the first wave only. However, the difference in proportions was not significant χ^2^(1) = 2, *p* = 0.157. Participants responding to both waves were significantly older with a mean of 47 years compared to a mean age of 44 years in the first wave, *t*(1386) = −4.96, *p* < 0.001. No significant differences were found in the study variable of burnout and off-job crafting between those who participated in both waves and those who participated in first wave only. The total paired sample was *N* = 658. The vast majority of the respondents were German (85%), whereas a minority of the sample was of Swiss nationality (15%). Most of the respondents finished their secondary school (63%), followed by tertiary (30%) and primary (7%) education. Concerning gender, about half of the sample were men (55%) and women (45%).

### 2.2. Measures

Work-Related Burnout. The seven-item measure from the original 19-item Copenhagen Burnout Inventory was used [25]. Items were answered on a five-point scale from 1 = “never/almost never” to 5 = “very often”. All items captured the degree of physical and psychological fatigue and exhaustion that is perceived by the person as related to their work; for example: “do you feel burnt out because of your work?”. One item was recoded so that a higher score indicates higher levels of burnout. The scale showed good reliability before (α = 0.89) and during (α = 0.87) the crisis.

Off-Job Crafting. The 18-item version of the Needs-based Off-job Crafting Scale (NOCS) was used to measure off-job crafting over the past month, comprising the following six dimensions: detachment, relaxation, autonomy, mastery, meaning, and affiliation [13]. All items measured the extent to which people restore their psychological needs in their non-working life on a five-point scale from 1 = “never” to 5 = “very often”. A sample item reads: “I have organized my free time in such a way that I switch off from professional duties”. The scale and its subdimensions showed excellent reliability before (α = 0.91) and during (α = 0.91) the crisis. Concerning the off-job crafting subdimensions, the scales showed excellent reliabilities given the low number of items before and during the crisis (see Table 1).

Demographic and COVID-19 Specific Variables. We measured key demographic variables, including age, gender, education, and nationality.

### 2.3. Statistical Analysis

Data analyses were conducted using the statistical software R [34]. Although we did not identify any outliers in our sample, we still inspected the extreme values in overall mean scores (i.e., 1 and 5). In total, there were 50 participants with at least one such “extreme” score. A more detailed analysis showed that there were no extremes scores that would be suspicious (i.e., suggesting mindless responding). Moreover, we implemented attention checks in the survey (e.g., “Please choose option 4”) to filter out random answers. Therefore, we did not exclude these values as these values were possible and they occurred only in a small number of cases.

Before conducting the analyses for the research questions, we first tested both concepts (i.e., burnout and off-job crafting) for configural, metric, and scalar invariance across the two time points using the full information maximum likelihood estimation procedure from the *lavaan* package [35]. To evaluate the models, we used comparative-fit index (CFI), root mean square error of approximation (RMSEA), and standardized root mean square residual (SRMR) with the conventional cut-off values. The goodness-of-fit values for CFI surpassing 0.90 indicated an acceptable fit and exceeding 0.95 indicated a good fit [36]. A value under 0.08 for SRMR and RMSEA indicated a good fit [37]. The models were compared using chi-square difference tests. The configural fit for OJC was good (CFI = 0.983; RMSEA = 0.04, SRMR = 0.03), and we then tested for metric invariance in which the factor loadings were constrained to be equal across the two time points. The model fit was good and there were no significant differences between the two models: Δχ^2^ = 9.9, Δdf = 11, *p* = 0.54. Finally, we tested for scalar invariance where the intercepts were also fixed to be equal across the two time points. The fit indices were good and the model did not significantly differ from the metric model: Δχ^2^ = 18.7, Δdf = 11, *p* = 0.07. The configural fit for burnout was good (CFI = 0.968; RMSEA = 0.09, SRMR = 0.03), so we proceeded with metric invariance. The metric fit was good and there were no significant differences between the two models: Δχ^2^ = 3.18, Δdf = 5, *p* = 0.67. Finally, we tested for scalar invariance. the fit indices were good and the model did not significantly differ from the metric model: Δχ^2^ = 5.06, Δdf = 5, *p* = 0.41. We were therefore able to establish configural, metric, and scalar invariance for both studied variables, enabling us to conduct appropriate analyses for the three research questions.

To examine the impact of the pandemic on off-job crafting and burnout (Research Question 1), pre–post comparisons were conducted using paired sample t-tests. Correlation analyses were used to assess the cross-sectional associations between off-job crafting and burnout before and during the COVID-19 crisis, respectively (Research Question 2a), and the longitudinal associations between off-job crafting before the crisis and burnout during the pandemic (Research Question 2b). Multiple hierarchical linear regression analyses were conducted to examine the predictive value of off-job crafting before the crisis on burnout during the crisis, controlling for burnout before the crisis (Research Question 3). By controlling for these variables in the second step, we were able to assess the predictive value of the off-job crafting dimensions before the crisis on a change in burnout levels during crisis, ruling out possible confounding effects of other variables affecting burnout at baseline. We conducted this regression analysis using both the total off-job crafting concept to test its explanatory power for burnout during the crisis as well as its six subdimensions to test which off-job crafting subdimensions predict the biggest change in burnout. Annotated R script with all the analysis is included as Appendix A.

### 2.4. Ethical Considerations

Informed consent was obtained from all participants. The study included adult participants (18+ years) only. Participants voluntarily completed the questionnaires, guaranteeing their anonymity. For anonymous surveys on working/living conditions and self-reported mental well-being and health, no ethical review was necessary under national (Swiss Human Research Act), university (Central Ethics Committee of the University of Zurich, https://www.research.uzh.ch/en/procedures/ethikkommissionen.html, accessed on 23 December 2021), or departmental rules (Department of Data Protection at the University of Zurich, www.dsd.uzh.ch/en/, accessed on 23 December 2021). The study was conducted under strict observation of ethical and professional guidelines.

## 3. Results

### 3.1. Changes and Associations between Off-Job Crafting and Burnout before and during the Crisis

Related to Research Question 1, Table 1 shows the means and the results of the bivariate correlation analyses between off-job crafting and burnout before the onset of the pandemic. The results of the paired-sample t-tests showed a significant decrease in burnout at T2 (Δ*M* = −0.10) compared to T1, *t*(657) = 3.91, *p* < 0.001, and in the dimension of off-job crafting for affiliation (Δ*M* = −0.16), *t*(649) = 4.78, *p* < 0.001; see Figure 1 and Figure 2. The effect size was small in both cases (*d* = 0.19; *d* = 0.15, respectively). The other off-job crafting dimensions (i.e., crafting for detachment, relaxation, autonomy, mastery, and meaning) did not significantly change between T1 and T2.

Concerning Research Question 2, the results showed that all subdimensions and the combined off-job crafting score (i.e., average across all dimensions) were cross-sectionally negatively associated with burnout before and during the COVID-19 crisis. These negative associations between off-job crafting and burnout were also present longitudinally, indicating that employees who off-job crafted for detachment, relaxation, autonomy, mastery, meaning, and affiliation before the crisis experienced fewer burnout complaints during the crisis. Off-job crafting for affiliation showed the strongest negative association with burnout, followed by off-job crafting for detachment, autonomy, mastery, meaning, and relaxation.

### 3.2. Predictive Values of Off-Job Crafting before the Crisis on Burnout during the Crisis

Table 2 shows the results of the multiple hierarchical regression analyses to address Research Question 3. The results of the regression analysis for Model 1 indicated that the model was a significant predictor of burnout during the crisis: R^2^ = 0.51, F(2, 655) = 341.4, *p* < 0.001 explaining 51% of the variance. Burnout during the crisis was significantly predicted by the total concept of off-job crafting before the crisis, meaning that employees who crafted before the crisis experienced less burnout during the crisis.

When looking at the subdimensions of off-job crafting in Model 2, off-job crafting before the crisis was a significant predictor of burnout during the crisis: R2 = 0.51, F(7, 650) = 100.3, *p* < 0.001, explaining 51% of the variance. Burnout was most strongly predicted by detachment followed by affiliation. Relaxation, autonomy, mastery, and meaning were not significant predictors. This suggests that employees who crafted for detachment and affiliation before the crisis experienced less burnout during the crisis.

The results of the regression analysis for Model 3 showed that off-job crafting before the crisis was a significant predictor of burnout during the crisis: R2 = 0.52, F(3, 650) = 233.4, *p* < 0.001, explaining 52% of the variance. This indicates that, when controlling for burnout and off-job crafting before the crisis, burnout during the crisis was predicted by off-job crafting during the crisis. Therefore, people who crafted during the crisis experienced fewer changes in burnout during the crisis. 

Looking at the subdimensions of off-job crafting in Model 4, the model showed that off-job crafting during the crisis was a significant predictor of burnout during the crisis: R2 = 0.52, F(7, 630) = 99.2, *p* < 0.001, explaining 52% of the variance. Burnout was significantly predicted by crafting for detachment, but none of the other subdimensions reached significance. This suggests that employees who crafted for detachment during the crisis experienced less burnout during the crisis.

## 4. Discussion

### 4.1. Theoretical Implications

The overarching aim of this study was to examine the role of off-job crafting in burnout prevention during the COVID-19 crisis. First, we synthesize the results of our longitudinal study, followed by practical implications, reflections on study limitations and strengths, and directions for future studies in this research area.

The results regarding Research Question 1 showed that both burnout and off-job crafting for affiliation decreased during the crisis compared to before the crisis. After the onset of the pandemic, employees experienced less physical and psychological fatigue and exhaustion related to their job. This is in line with Kimhi et al. [38], who showed that negative outcomes such as distress symptoms, decreased, while positive outcomes such as perceived well-being, increased during the crisis. Since burnout is a work-related phenomenon [20], the decrease in burnout levels may be attributed to the slowing down of the whole economy due to the crisis, which potentially reduced experienced job demands such as workload and increased job resources such as autonomy. Speculatively, the crisis may have ‘forced’ employees to also slow down their everyday pace of life, thereby supporting their recovery from work in their non-working context. Related to this, the decrease in off-job crafting for affiliation may be attributed to the lockdown, in particular the social distancing measures, which significantly reduced the opportunities for socializing with others.

The results concerning Research Questions 2a and 2b indicate that, even though burnout and off-job crafting decreased during the crisis, employees who crafted in relation to detachment and affiliation experienced less burnout before and during the crisis. Although previous studies have not yet focused on a differentiated perspective for the six off-job crafting dimensions for burnout prevention, the finding aligns with previous studies investigating the benefit of satisfaction of the six DRAMMA dimensions in relation to work-related stress [13,20]. The results also align with studies focusing on other coping resources, such as having a positive attitude, which has shown to mediate the relationship between perceived stress and life satisfaction during the pandemic [39]. The present study complements this research by showing that the overall off-job crafting concept and its subdimensions are all negatively associated with burnout both cross-sectionally and longitudinally in crisis situations. Moreover, the results related to Research Questions 3a and 3b showed that people who off-job crafted for specific crafting dimensions before and during the crisis reported less burnout during the crisis. In this regard, looking at the off-job crafting dimensions, employees who crafted for detachment before and during the crisis and for affiliation before the crisis reported fewer burnout complaints during the crisis. Our results indicate that detachment seems to be the most relevant dimension of off-job crafting in relation to burnout. This is in line with previous research that indicates the protective value of detachment against burnout [13]. Employees that crafted their off-job time according to this specific need might consequently be better able to detach from work during the crisis. Affiliation is also known to be a protective resource against burnout in the workplace [40], which indicates that even in times of a rigorous lockdown employees could still benefit from the experience of affiliating in terms of lower risk of burnout complaints. The other off-job crafting dimensions (i.e., relaxation, autonomy, mastery, and meaning) did not significantly predict burnout in any of the regression models, suggesting that these dimensions do not play a big role in burnout prevention during crisis.

### 4.2. Practical Implications

The present findings yield valuable insights for practice (e.g., interventions) to stimulate off-job crafting behaviors, which may in turn prevent the onset or alleviate burnout complaints. The results showed that crafting in relation to detachment and affiliation seems particularly important in burnout prevention in crisis situations. Both of these crafting dimensions can possibly be targeted through interventions on recovery in general [41] and by a recently designed hybrid off-job crafting intervention [42]. With regard to recovery interventions in general, Hahn et al. [41] developed and evaluated a recovery training program comprising multiple training sessions targeting employees’ control during off-job time, psychological detachment from work, transition rituals, mastery experiences, and relaxation and sleep. The effect evaluation of this intervention showed that the intervention decreased perceived stress, among other outcomes [41].

Concerning the above proposed hybrid off-job crafting intervention, Kosenkranius et al. [42] developed two on-site group training sessions in which employees learn about off-job crafting and how to achieve the satisfaction of needs. Additionally, a smartphone app (i.e., Everydaily) has been developed to support employee engagement in off-job crafting. Participants receive daily suggestions for three different activities to help them satisfy their psychological needs, such as engaging in nature walks, mindfulness, volunteer work, or scheduling an hour of “me-time” in the agenda. Although the intervention has not been evaluated yet regarding its effectiveness, it may inspire companies and provide them with ideas to stimulate off-job crafting behaviors. Implementing such interventions might also be relevant for public health policies during this COVID-19 pandemic.

### 4.3. Limitations and Strengths

This study has several limitations and strengths that should be taken into account when interpreting the results, which we translated into possible directions for future studies. It is a particular strength of this study that it refers to a broad and relatively large sample, offering a pre–post study design concerning the COVID-19 crisis. A first limitation is that all measures were self-reported, which may bring multiple, common methodological biases, such as socially desirable answers, aggravated by the length of the survey [43]. However, we used valid and reliable scales to enhance the internal validity of the study, as reflected in the good reliability of all scales. It is a strength of this study that it applied a longitudinal study design, using one wave before the crisis and one wave during the pandemic. This allowed us to comprehensively test for changes in burnout and off-job crafting during the crisis compared to before the crisis and to examine the extent to which off-job crafting before and during the crisis changed burnout levels during the crisis, covering one year and similar periods of the year (March 2019–April 2020). Future research may identify how off-job crafting across even shorter time intervals (e.g., daily or weekly) exerts its influence on the prevention of burnout [44] and how daily crafting can be promoted through interventions [42]. Second, it is of interest to examine the role of off-job crafting in relation to restoring employees’ ability to regulate cognitive and emotional processes, as an impediment of these processes has recently been acknowledged as additional burnout symptoms [45]. Being physically and mentally exhausted still remain important core symptoms of burnout (i.e., “I can no longer do my job”). However, these symptoms intertwined with a reduced capacity to regulate cognitive and emotional functioning (i.e., “I do not want to do my job anymore”), and are often accompanied by a depressed mood and non-specific psychological and psychosomatic tension complaints [45,46]. The present study could not consider these recent developments in research, as we started collecting data long before the publication of these recent measures and manuals for burnout research. Future studies are hence encouraged to understand how employees’ crafting styles in their non-working life can potentially also reduce burnout through enhancing or restoring cognitive and emotional functioning.

Third, although the findings of the present study show that off-job crafting contributes to fewer burnout complaints during the crisis, the explained variance and standardized beta coefficients were relatively small. This implies that other factors may play a role in burnout prevention in crisis situations, for instance, the role of job security, family relationships, proactive personality, and actually contracting the corona disease. Future studies are recommended to include such factors. Moreover, 85% of the participants in our study were German, which means that the results should be generalized with caution to the underrepresented Swiss population or other countries.

Finally, it could be that employees with low levels of burnout complaints are able to craft more proactively than those with high levels of burnout complaints (i.e., reverse causation). However, we controlled the regression analyses for burnout before the crisis and the results do suggest that employees who proactively crafted before the crisis experienced less burnout during the crisis—independent of their baseline level of burnout. Nevertheless, future studies are encouraged to test possible reverse causations between off-job crafting and burnout within and beyond crisis situations. We also tested whether burnout at T1 could predict changes in off-job crafting between T1 and T2 and did not find statistically significant relationships. This suggests that the relationship between off-job crafting and burnout is in the expected (causal) direction.

## 5. Conclusions

The present study shows that both burnout and off-job crafting for affiliation decreased during the COVID-19 crisis compared to before the crisis, while other off-job crafting dimensions remained stable across time. The findings also show that employees who crafted in their off-job time before the crisis experienced fewer burnout complaints during the crisis. Looking more closely at the subdimensions of off-job crafting, employees who crafted for detachment and affiliation before the crisis, and those crafting for detachment during the crisis, reported less burnout during the crisis. Overall, the present study offers unique insights into how employees can proactively craft during crisis situations and complements the existing body of knowledge on burnout prevention. We hope these findings will encourage future researchers to examine the role of off-job crafting in burnout prevention beyond crisis situations and help to develop interventions to increase employees’ off-job crafting capacities.

## Figures and Tables

**Figure 1 ijerph-19-02146-f001:**
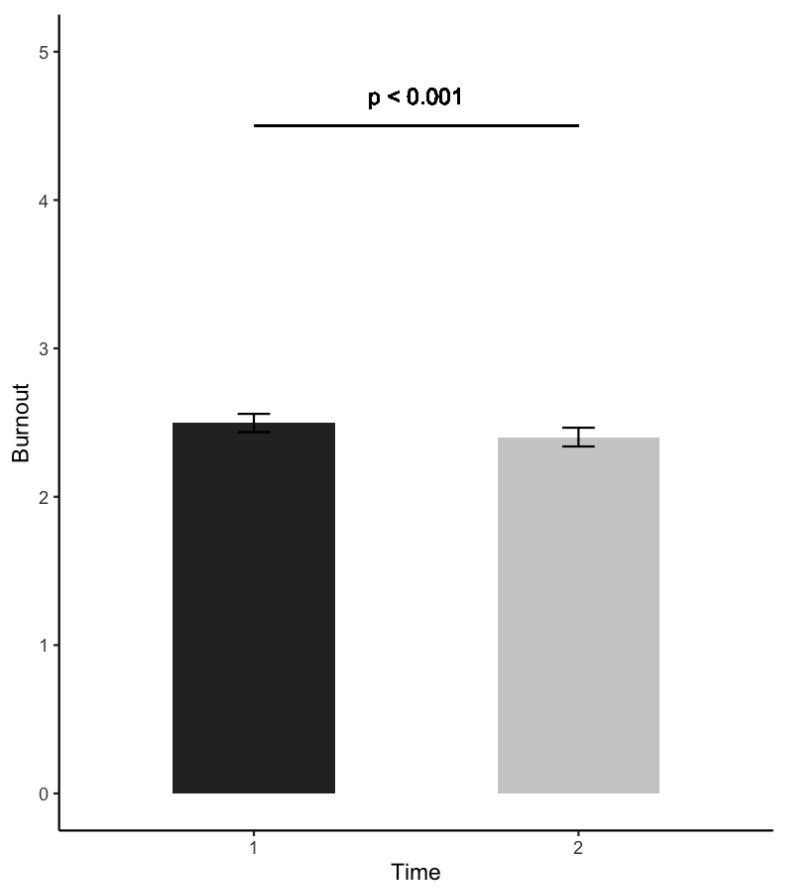
Pre-test (before the crisis) and post-test (during the crisis) changes in burnout.

**Figure 2 ijerph-19-02146-f002:**
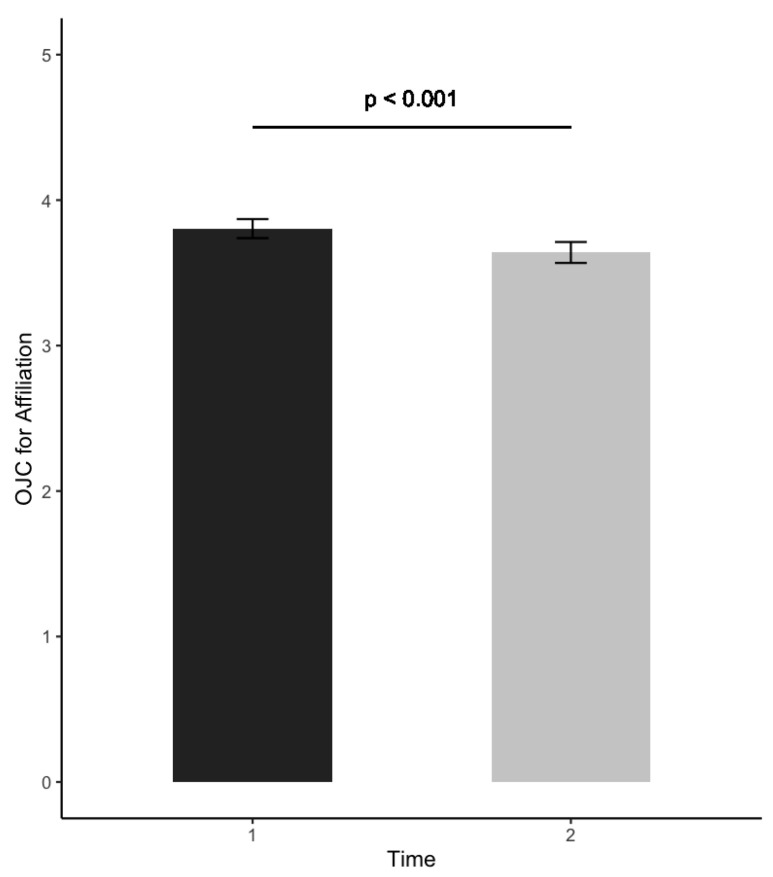
Pre-test (before the crisis) and post-test (during the crisis) changes in off-job crafting (OJC) for affiliation.

**Table 1 ijerph-19-02146-t001:** Means, Standard Deviations, Cronbach Alphas, and Bivariate Correlations between the Study Variables.

	*M*	*SD*	*α*	1	2	3	4	5	6	7	8	9	10	11	12	13	14	15	16
1. Burnout T1	2.50	0.80	0.89	1															
2. Burnout T2	2.40	0.83	0.87	0.71	1														
3. OJC T1	3.74	0.61	0.91	−0.27	−0.29	1													
4. OJC T2	3.71	0.65	0.91	−0.29	−0.24	0.59	1												
3. OJC for Detachment T1	3.91	0.88	0.86	−0.19	−0.24	0.69	0.41	1											
4. OJC for Relaxation T1	3.79	0.83	0.83	−0.17	−0.17	0.73	0.43	0.66	1										
5. OJC for Autonomy T1	3.85	0.76	0.75	−0.23	−0.22	0.80	0.48	0.45	0.52	1									
6. OJC for Mastery T1	3.44	0.87	0.82	−0.18	−0.20	0.76	0.46	0.30	0.39	0.63	1								
7. OJC for Meaning T1	3.61	0.79	0.78	−0.21	−0.20	0.77	0.46	0.28	0.33	0.57	0.70	1							
8. OJC for Affiliation T1	3.80	0.86	0.89	−0.25	−0.28	0.71	0.40	0.33	0.33	0.45	0.44	0.62	1						
9. OJC for Detachment T2	3.93	0.89	0.85	0.08 *	−0.18	0.38	0.70	0.50	0.36	0.28	0.16	0.19	0.19	1					
10. OJC for Relaxation T2	3.82	0.82	0.80	−0.09 *	−0.15	0.42	0.73	0.34	0.47	0.37	0.25	0.25	0.21	0.62	1				
11. OJC for Autonomy T2	3.82	0.83	0.80	−0.11 **	−0.17	0.47	0.83	0.32	0.35	0.47	0.38	0.35	0.26	0.50	0.57	1			
12. OJC for Mastery T2	3.45	0.87	0.82	−0.14	−0.21	0.47	0.78	0.22	0.27	0.38	0.53	0.45	0.29	0.36	0.42	0.66	1		
13. OJC for Meaning T2	3.57	0.81	0.76	−0.13 **	−0.19	0.48	0.77	0.20	0.23	0.41	0.48	0.52	0.34	0.32	0.37	0.61	0.66	1	
14. OJC for Affiliation T2	3.64	0.93	0.89	−0.13 *	−0.19	0.44	0.70	0.24	0.24	0.30	0.30	0.37	0.51	0.32	0.33	0.43	0.48	0.58	1

*Note. N* = 658; all correlations are significant at *p* < 0.001, if not indicated otherwise; ** *p* < 0.01 * *p* < 0.05.

**Table 2 ijerph-19-02146-t002:** Multiple Linear Regression Analyses between Off-Job Crafting and Burnout during Crisis (T2).

Model	Predictor	Estimate	SE	95% CI		β	R^2^	F(df)	ΔR^2^	ΔF
				LL	UL					
*Model 1*										
Step 1	Burnout T1	0.73	0.03	0.67	0.78	0.71 ***	0.498	653.5(1, 656)		
Step 2	Burnout T1	0.69	0.03	0.64	0.75	0.68 ***	0.508			
	OJC T1	0.14	0.04	−0.22	−0.07	−0.11 ***				
								341.4(2, 655)	0.010	312.1 ***
*Model 2*										
Step 1	Burnout T1	0.73	0.03	0.67	0.78	0.71 ***	0.498	653.5(1, 656)		
Step 2	Burnout T1	0.69	0.03	0.63	0.75	0.67 ***				
	OJC for Detachment T1	−0.12	0.18	−0.19	−0.05	−0.13 ***				
	OJC for Relaxation T1	0.06	0.03	−0.02	0.13	0.06				
	OJC for Autonomy T1	0.01	0.04	−0.07	0.09	0.01				
	OJC for Mastery T1	−0.07	0.04	−0.15	0.01	−0.07				
	OJC for Meaning T1	0.04	0.04	−0.04	0.13	0.04				
	OJC for Affiliation T1	−0.07	0.03	−0.14	−0.01	−0.07 *				
							0.514	100.3(7, 650)	0.016	553.2 ***
*Model 3*										
Step 1	Burnout T1	0.69	0.03	0.64	0.75	0.68 ***	0.508	341.4(2, 655)		
	OJC T1	−0.14	0.04	−0.22	−0.07	−0.11 ***				
Step 2	Burnout T1	0.69	0.03	0.64	0.75	0.68 ***				
	OJC T1	−0.05	0.05	−0.14	0.04	−0.04				
	OJC T2	−0.16	0.04	−0.24	−0.07	−0.12 ***				
							0.516	233.4(3, 650)	0.008	108 ***
*Model 4*										
Step 1	Burnout T1	0.73	0.03	0.67	0.78	0.71 ***	0.498	653.5(1, 656)		
Step 2	Burnout T1	0.69	0.03	0.63	0.75	0.67 ***				
	OJC for Detachment T2	−0.09	0.03	−0.16	−0.03	−0.10 **				
	OJC for Relaxation T2	0.01	0.04	−0.07	0.08	0.01				
	OJC for Autonomy T2	0.02	0.04	−0.06	0.10	0.02				
	OJC for Mastery T2	−0.05	0.04	−0.12	0.03	−0.05				
	OJC for Meaning T2	−0.03	0.04	−0.11	0.05	−0.03				
	OJC for Affiliation T2	−0.03	0.03	−0.09	0.03	−0.04				
							0.519	99.2(7, 630)	0.021	554.3 ***

*Note. N* = 658; *** *p* < 0.001 ** *p* < 0.01 * *p* < 0.05.

## Data Availability

Data will be made available based on reasonable request.

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
