# Peer review of "The Role of Off-Job Crafting in Burnout Prevention during COVID-19 Crisis: A Longitudinal Study"

_ijerph, 2022, doi:10.3390/ijerph19042146_

Round 1
Reviewer 1 Report
This study examined the relationship between off-job crafting and burnout across the COVID-19 crisis using a longitudinal research design. Throughout the manuscript, an adequate literature review is carried out, the statistical analysis being consistent with the theoretical framework and correctly executed.
However, regarding the analysis, some more detail is required, e.g. the authors mention that the total paired sample was N = 658 – but not if (or rather how many) outliers were deleted, what the procedure for this was etc. Also, effect size of the paired-sample t-tests should be provided in line 238-241.
Reviewer 2 Report
The study’s theme is very interesting, and the manuscript is well written. In my opinion, this is almost ready for publication. I only have some minor doubts that I would like to clarify.
Abstract
Pag. 1, line 17: In the abstract it is reported that the sample of 658 participants is made up of German and Swiss employees: it is better to report already in the abstract Mean, SD and percentage of males or females.
Method
Pag. 4, line 176: In the participants section there is no information about the sample, such as eg. the percentage of males and females, the mean and standard deviation of the sample, the percentage of how many participants are from the German and how many from the Swiss. These data are reported in the results section, but I think they are more explanatory in the method part.
Pag. 4, line 176: In the part of the method there should be more information about ethical issue. Has this research project been accepted by the ethical commission? Were the participants informed about the research aims? Were the participants informed that their participation in the study was voluntary? It would be better to specify the whole procedure.
Pag. 4, line 231: The image of the table is very small, not of good image quality. They need to be fixed.
Results
Pag. 4, line 198: It is reported that the Copenhagen Burnout Inventory "showed excellent reliability before (α = .89) and during (α = .87) the crisis". Unless there are other bibliographic sources, the value below 0.90 is not "excellent", but "good". Carrying over of the bibliography, to be noted in the paper, which supports these values: George, D. (2011). SPSS for windows step by step: A simple study guide and reference, 17.0 update, 10/e. Pearson Education India.
Discussion
Pag. 7, line 279: I think it can be useful not only to show your results, but also to refer to the international literature and a brief comparison with other works. For this I recommend an integration based on the following ideas (not exhaustive):
- Angelini, G., Buonomo, I., Benevene, P., Consiglio, P., Romano, L., & Fiorilli, C. (2021). The Burnout Assessment Tool (BAT): A contribution to Italian validation with teachers’. Sustainability, 13(16), 9065.
- Brooks, S. K., Webster, R. K., Smith, L. E., Woodland, L., Wessely, S., Greenberg, N., & Rubin, G. J. (2020). The psychological impact of quarantine and how to reduce it: rapid review of the evidence. The lancet, 395(10227), 912-920.
- Fiorilli, C., Pepe, A., Buonomo, I., & Albanese, O. (2017). At-risk teachers: the association between burnout levels and emotional appraisal processes. The Open Psychology Journal, 10(1).
- Gori, A., Topino, E., & Di Fabio, A. (2020). The protective role of life satisfaction, coping strategies and defense mechanisms on perceived stress due to COVID-19 emergency: A chained mediation model. Plos one, 15(11), e0242402.
- Gori, A., & Topino, E. (2021). Across the COVID-19 Waves; Assessing Temporal Fluctuations in Perceived Stress, Post-Traumatic Symptoms, Worry, Anxiety and Civic Moral Disengagement over One Year of Pandemic. International Journal of Environmental Research and Public Health, 18(11), 5651.
- Pedersen, M. J., & Favero, N. (2020). Social distancing during the COVID‐19 pandemic: who are the present and future noncompliers?. Public Administration Review, 80(5), 805-814.
- Pyhältö, K., Pietarinen, J., Haverinen, K., Tikkanen, L., & Soini, T. (2020). Teacher burnout profiles and proactive strategies. European Journal of Psychology of Education, 1-24.
- Kniffin, K. M., Narayanan, J., Anseel, F., Antonakis, J., Ashford, S. P., Bakker, A. B., ... & Vugt, M. V. (2021). COVID-19 and the workplace: Implications, issues, and insights for future research and action. American Psychologist, 76(1), 63.
Pag. 9, line 348: I believe that the fact that the sample is for the most part (85%) made up of Germans should be placed within the limits of the study.
Reviewer 3 Report
Overall, I found this manuscript to be very well written and to address a timely and relevant topic. There are, however, a few issues to be considered which I have outlines below.
- The following paper provides a relevant addition to how COVID-19 affected employee burnout experiences through different attitudes and appears helpful to strengthen the introduction further: https://doi.org/10.1016/j.chb.2020.106677
- I found the first section in 1.1 to be difficult to follow. Breaking to down further and remove unnecessary details (would appear helpful to me.
- The research questions are well derived from the literature and interesting. I was confused why the authors did not formulate hypotheses. In the theoretical background they compellingly proposed directed relationships that they expected. Thus I would have expected to read such hypotheses here too.
- A central premise for comparing T1 and T2 data is measurement invariance. This still needs to be established before being able to compare the means at both measurement points.
- Conducting the analyses on a latent level would seem helpful to me, especially as some of the internal consistencies are fairly low (e.g., OJC).
- A figure documenting the changes from T1 and T2 would be helpful for better interpretability of the results.
- How much missing data was there and how was this handled? How many participants participated at T1, how many at T2, and how was drop-out analyses (systematic drop out analyses for instance)?
- A statement on the availability of the data and the analytical script should be added to ensure reproducibility of results by other researchers.
Round 2
Reviewer 3 Report
The authors have adequately addressed the issues that I raised and substantially improved the manuscript. I congratulate them to what I deem a successful revision.